# Dual-Functional Peroxidase-Copper Phosphate Hybrid Nanoflowers for Sensitive Detection of Biological Thiols

**DOI:** 10.3390/ijms23010366

**Published:** 2021-12-29

**Authors:** Xuan Ai Le, Thao Nguyen Le, Moon Il Kim

**Affiliations:** Department of BioNano Technology, Gachon University, 1342 Seongnamdae-ro, Sujeong-gu, Seongnam 13120, Gyeonggi, Korea; xuanai.le6667@gmail.com (X.A.L.); thaonguyen65949@gmail.com (T.N.L.)

**Keywords:** hybrid nanoflowers, fluorescent biosensors, biological thiols, thiol oxidase mimic, cascade reaction

## Abstract

An effective strategy to detect biological thiols (biothiols), including glutathione (GSH), cysteine (Cys), and homocysteine (Hcy), holds significant incentive since they play vital roles in many cellular processes and are closely related to many diseases. Here, we demonstrated that hybrid nanoflowers composed of crystalline copper phosphate and horseradish peroxidase (HRP) served as a functional unit exhibiting dual catalytic activities of biothiol oxidase and HRP, yielding a cascade reaction system for a sensitive one-pot fluorescent detection of biothiols. The nanoflowers were synthesized through the anisotropic growth of copper phosphate petals coordinated with the amine/amide moieties of HRP, by simply incubating HRP and copper(II) sulfate for three days at room temperature. Copper phosphates within the nanoflowers oxidized target biothiols to generate H_2_O_2_, which activated the entrapped HRP to oxidize the employed Amplex UltraRed substrate to produce intense fluorescence. Using this strategy, biothiols were selectively and sensitively detected by monitoring the respective fluorescence intensity. This nanoflower-based strategy was also successfully employed for reliable quantification of biothiols present in human serum, demonstrating its great potential for clinical diagnostics.

## 1. Introduction

Thiol group-containing amino acids, such as glutathione (GSH), cysteine (Cys), and homocysteine (Hcy), termed as biological thiols (biothiols), play significant roles in various physiological and pathological processes [1]. In human body, biothiols serve as major antioxidants to reduce oxidative damage from free radicals, inhibit cellular apoptosis, and control the three-dimensional structure of protein molecules [2,3,4]. Thus, the abnormal level of biothiols is closely related to many health problems. For example, decreased GSH level is associated with declined activity in immune and nervous systems, as well as impairment of antioxidant defenses, while its elevation may indicate the progression of cancer cells during chemotherapy or radiotherapy [5,6,7]. Cys depletion is affiliated to edema, hair depigmentation, muscle atrophy, and skin lesions, while its increased level correlates with neurotoxicity [8]. Elevated Hcy level represents homocystinuria, which can cause many serious diseases such as atherosclerosis, Alzheimer’s disease, and cardiovascular diseases [9,10]. Furthermore, the level of total biothiols can be used as a reliable biomarker for monitoring the oxidative stress, particularly for predicting the survival rates of oral cancer patients [11]. Therefore, reliable quantitative determination of individual biothiol as well as total biothiols is of great importance for early-stage diagnosis of the diseases and monitoring the relevant health states.

Instrumentation-based analytical techniques, such as high-performance liquid chromatography (HPLC) [12,13], capillary electrophoresis [14], and mass spectrometry [15], have been conventionally utilized for biothiol quantification. These methods enable reliable and sensitive determination; however, their utilization is frequently limited in many diagnostic circumstances, particularly in point-of-care testing (POCT) environments where analytical resources are not sufficiently provided. In this regard, biosensing strategy primarily based on the unique properties of nanomaterials has attracted intensive attention since it enables rapid, sensitive, and convenient identification of target biothiols without the involvement of any complex instrumentation or skilled operator [16]. Most of the reported biothiol biosensors utilized the distinct interaction involving the sulfhydryl group of biothiols. For example, inspired by the quenching of the fluorescence of gold nanoclusters from the coordination with mercury ions, a unique turn-on sensing strategy for biothiols was reported. By adding biothiols on a mercury ion-gold nanoclusters hybrid, mercury ions would be detached in order to prepare mercury-sulfur bond, consequently restoring the fluorescence of gold nanoclusters [17]. Polydiacetylene-based colorimetric/fluorometric dual-mode biosensing for biothiols was also reported. Due to the high affinity of pyridine-containing polydiacetylene toward sulfur, biothiols promoted the decomposition of the pyridine-mercury complex in the polydiacetylene, yielding a blue-to-red transition and concomitant fluorescence change [18]. Biothiols were also quantitatively determined based on the anti-etching of silver nanoprisms. Since biothiols bound to the surface of silver nanoprisms via the silver-sulfur interaction, protecting the silver nanoprisms from being etched by chloride ions, thus exhibiting the purple color. In the absence of biothiols, the color of silver nanoprisms turned to yellow through the respective etching. This strategy could also discriminate the kinds of biothiols since each biothiol induced different color transition by tuning the pH condition [19]. Although these examples demonstrate the potential of employing nanostructured materials for biothiol determination, it is still highly desirable to develop an advanced biosensing strategy, that enables more simple/rapid/robust to perform and sensitive/reliable/stable to determine the target biothiols.

Recently, organic-inorganic hybrid nanoflowers have gathered a growing attention in various fields due to their capability to efficiently entrap the organic components within the flower-like hierarchically structured matrices, which yielded not only highly retained activity but also improved stability [20,21]. They are easily synthesized under mild conditions, from the coordination between amine/amide moieties of organic components and inorganic ions including copper, zinc, manganese, cobalt, and iron, followed by anisotropic growth to form highly branched flower-like morphologies. Various enzymes were incorporated within the nanoflowers to achieve improved catalytic properties and extensively applied to biosensor, bioremediation, biofuel cell, and other bioconversions [22,23,24]. However, the nanoflowers have not been widely investigated to seek their enzyme-mimicking activities, except for several peroxidase and laccase-mimicking cases [25,26].

Herein, we demonstrated that the hybrid nanoflowers composed of crystalline copper phosphate and protein molecules showed oxidase-like activity toward biothiols. Based on this observation, we confirmed that the hybrid nanoflowers composed of horseradish peroxidase (HRP) and copper phosphate (HRP-Cu NFs) possessed dual enzymatic activities, which were successfully utilized to construct a cascade reaction system to detect biothiols. Various analytical characteristics of the HRP-Cu NFs-based system, such as selectivity, sensitivity, stability, and detection precisions, were investigated.

## 2. Results and Discussion

### 2.1. Dual Catalytic Activities of HRP-Cu NFs for the Detection of Biothiols

The synthesis of HRP-Cu NFs and their applications to biothiol detection are illustrated in Figure 1. Copper elements, which might play an essential role in biothiol oxidation, formed complexes with amine/amide moieties of HRP by coordination interaction, which served as a nucleation site for primary copper phosphate crystals. Thereafter, flower petals that incorporate crystalline copper phosphate and HRP were anisotropically grown, consequently yielding flower-like nanostructured microparticles during three days of incubation at room temperature (RT). Previous works demonstrated that the enzyme-inorganic hybrid nanoflowers showed high activity retention and improved stability, based on the affirmative confinement of enzyme molecules within the flower matrices [20]; however, their enzyme-mimicking activity arisen from the crystalline inorganic phosphate has not been widely studied. We hypothesized that copper phosphate crystals present within the nanoflowers could show biothiol oxidase-like activity, and thus, HRP-Cu NFs could have dual catalytic activities, enabling cascade reaction to detect biothiols. Specially, in the presence of target biothiols, the biothiol oxidase-like action is expected to generate H_2_O_2_, which should activate HRP entrapped within the nanoflowers to convert employed Amplex UltraRed reagent (AUR) into highly fluorescent product.

Flower-like hierarchically structured shape of HRP-Cu NFs was confirmed by SEM images with a diameter of approximately 20 μm (Appendix A). The encapsulation yield of HRP was calculated to be ~90%, which was comparable with the previously reported values [20]. HRP-Cu NFs were composed of Cu, P, O, and N elements, which might be from copper phosphate and HRP (Appendix A). In the Fourier transform infrared (FT-IR) spectra (Appendix A), the appearance of peaks for HRP-Cu NFs and Cu_3_(PO_4_)_2_ precipitate at 3430 cm^−^^1^ indicated the existence of O-H vibration. Sharp peaks at 1140 and 1040 cm^−^^1^, observed for both HRP-Cu NFs and Cu_3_(PO_4_)_2_, were attributed to P=O and P-O linkages, demonstrating the presence of phosphate groups. The peaks at 621 and 559 cm^−^^1^ represented metal-oxygen bonding, revealing the occurrence of Cu-O bond. X-ray diffraction (XRD) patterns of both HRP-Cu NFs and Cu_3_(PO_4_)_2_ were in good agreement with those of standard JCPDS (00-022-0548), corresponding to crystalline Cu_3_(PO_4_)_2_∙3H_2_O (Appendix A), although the peak intensities of HRP-Cu NFs decreased possibly due to the presence of embedded HRP. Average crystallite sizes of HRP-Cu NFs and Cu_3_(PO_4_)_2_ precipitates were calculated using the Scherrer equation: D = (K λ/β cosθ), where D is the crystallite size (nm), K is the Scherrer constant, λ represents the wavelength of X-ray, β is the full width at half maximum (FWHM) of the peak in radians, and θ is the diffraction angle in radians [27]. Accordingly, the average sizes of the crystallites in HRP-Cu NFs and Cu_3_(PO_4_)_2_ were found to be 22.2 and 45.9 nm, respectively. All these characterizations confirmed that HRP-Cu NFs were successfully formed during the mild synthetic condition.

Cascade reaction activity of HRP-Cu NFs, composed of HRP and expected biothiol oxidase, was evaluated by performing biothiol detecting reactions using AUR substrate, and the fluorescent responses were monitored. The experimental results clearly show that the HRP-Cu NFs enabled the detection of GSH, Cys, and Hcy from the oxidation of AUR to produce intense red fluorescence (Figure 2a). Without the addition of biothiols, negligible signals were detected. The catalytic behaviors of HRP-Cu NFs towards GSH, Cys, and Hcy were further investigated by comparing their biothiol detecting activity with that of individual catalytic components such as free HRP or bovine serum albumin-copper nanoflowers (BSA-Cu NFs) (Figure 2b–d). As expected, significantly high fluorescent signals were recorded when HRP-Cu NFs were employed with all three biothiols. No significant signal was observed when we employed free HRP only, due to the absence of biothiol oxidase activity. Interestingly, BSA-Cu NFs showed slight fluorescent signal towards biothiols. It was presumably due to their biothiol oxidase-mimicking activity, as well as marginal peroxidase-mimicking activity as reported recently [25] from the crystalline copper phosphate.

To confirm the biothiol oxidase-mimicking activity, further experiments were carried out to demonstrate the generation of H_2_O_2_ during the HRP-Cu NFs-mediated oxidation of biothiols by adding catalase (0.1 mg/mL) to the reaction mixtures, since catalase induces the conversion of H_2_O_2_ into H_2_O. As expected, significantly decreased fluorescent signals were observed with the addition of catalase for all three biothiols (GSH, Cys, and Hcy), demonstrating that H_2_O_2_ was produced during the HRP-Cu NFs-mediated oxidation of biothiols (Appendix A). The biothiol oxidase-like activity of HRP-Cu NFs can be ascribed to the Cu(II) present on their surfaces [28]. It is also important to verify that the biothiol detecting activity of HRP-Cu NFs results from the intact NFs and not from the free leaching copper ions. To confirm this, the GSH-detecting activity of HRP-Cu NFs was compared with that of the supernatant solution after separation of HRP-Cu NFs. The experimental results showed that a negligible fluorescent signal was produced from the supernatant, confirming that the catalytic activity was not associated with a leached ion but the intrinsic properties of the NFs (Appendix A) [28]. We next compared the biothiol detecting activity of HRP-Cu NFs with that of the control systems, which comprised of non-integrated free HRP with either BSA-Cu NFs or Cu_3_(PO_4_)_2_ precipitate. The investigation showed that HRP-Cu NFs had higher biothiol sensing activity, which was ~1.5-fold higher than that of the control system composed of free HRP with BSA-Cu NFs and ~2-fold higher than that of HRP with Cu_3_(PO_4_)_2_ precipitate (Appendix A). These results indicate that the incorporation of both free HRP and crystalline copper phosphate, located in close proximity of the nanoflowers, is advantageous to achieve high catalytic performance of cascade reaction for biothiol detection, as previously demonstrated for glucose detection [21]. The close distance of them enabled H_2_O_2_ produced from copper phosphate-catalyzed biothiol oxidation to be enriched around HRP, consequently boosting the cascade reaction activity for target biothiol detection.

### 2.2. Optimization of Reaction Conditions

We then optimized the reaction conditions for HRP-Cu NFs-mediated biothiol detecting by examining the effects of diverse reaction factors, such as pH, temperature, reaction time, and the employed concentrations of HRP-Cu NFs and AUR (Appendix A). The one-step cascade reaction was efficient under neutral pH, which is advantageous to detect samples in physiological pH conditions. Optimal temperature condition was placed around 40–50 °C, and we adopted 37 °C for further sensing experiments because of practical convenience and sufficiently acceptable catalytic performance of HRP-Cu NFs at 37 °C (about 90% of maximal activity). In addition, the fluorescence intensity showed a significant increase upon increasing the reaction time and the concentrations of HRP-Cu NFs and AUR, and it then remained almost constant beyond 15 min, 0.1 mg/mL, and 25 μM, which were selected and used for further experiments.

### 2.3. Analytical Capabilities of HRP-Cu NFs for the Detection of Biothiols

The feasibility of the HRP-Cu NFs-based system to detect biothiols was demonstrated. Through a typical one-pot reaction, target biothiols such as GSH, Cys, and Hcy were selectively detected by the generation of respective fluorescence (Figure 3). On the other hand, negative control samples such as carbohydrates (glucose, galactose, and maltose) and 15 different amino acids (alanine, arginine, asparagine, glycine, histidine, leucine, lysine, methionine, phenylalanine, proline, serine, threonine, tryptophan, tyrosine, and valine) did not generate any significant signal, even though they were used at 20-fold higher concentrations. Interestingly, Cys yielded higher fluorescence than that of GSH and Hcy, presumably due to its smaller size that may enable more efficient oxidation by HRP-Cu NFs. This result clearly demonstrates that the HRP-Cu NFs-based system shows excellent selectivity to the corresponding biothiols.

Quantitative analyses of GSH, Cys, and Hcy were performed through a typical 15-min reaction, followed by monitoring the respective fluorescence. The results showed that the fluorescence intensities versus GSH, Cys, and Hcy were linear up to 1 μM (Figure 4). The limit of detections (LODs) for GSH, Cys, and Hcy were calculated as low as 13.4, 4.5, and 18.3 nM, respectively, according to the equation LOD = 3 S/K [29,30], where S is the standard deviation of fluorescence for blank and K is the slope of the calibration plot. These LOD and the linear range values are among the best results of those recently reported biothiol sensors (Appendix A), revealing high sensitivity of the present system.

Entrapped HRP within the nanoflowers are expected to be more stable than the non-integrated free counterpart. To examine this, HRP-Cu NFs and a control free system containing BSA-Cu NFs and free HRP were incubated at pH 2–12 or 4–80 °C for 2 h, and then their GSH-detecting activities were evaluated. In accordance with the previous results of enzyme-inorganic hybrid nanoflowers [20], HRP-Cu NFs were found to maintain most of their activity at a wide range of pH or temperature, whereas the free HRP-based system lost most of its activity at an extreme pH or temperature over 50 °C (Appendix A). These observations clearly demonstrate high stability of HRP-Cu NFs, which is quite advantageous to realize practical applications.

Finally, we evaluated the clinical diagnostic utility of the HRP-Cu NFs-based sensor system by using real human blood serum samples that contained diverse levels of GSH, Cys, and Hcy. Considering our linear detection range (0.1–1 μM) and actual total biothiol level in human serum (400–600 μM) [31], human serum was 1000-fold diluted and then a predetermined amount of GSH, Cys, and Hcy (0.125, 0.25, and 0.5 μM) was further added to make spiked samples at diverse levels. As a result, the serum biothiols such as GSH, Cys, and Hcy were quantified with excellent precisions yielding CVs (coefficient of variation) in a range of 0.1–2.2% and recovery rates of 95.3–101.8% (Table 1), verifying the excellent reproducibility and reliability of the method. When the serum was diluted at a lower dilution ratio of 500-fold, a considerable reduction in recovery was observed due to higher GSH level (>1 μM) beyond our linear range (Appendix A). Thus, 1000-fold dilution is necessary to achieve the excellent analytical performance for biothiol detection in human serum using our proposed method. These results show that the HRP-Cu NFs-based biosensing system should serve as a promising analytical platform to diagnose total biothiols in clinical settings.

## 3. Materials and Methods

### 3.1. Materials

GSH, Cys, Hcy, copper (II) sulfate pentahydrate, HRP, phosphate buffered saline (PBS), sodium phosphate monobasic (NaH_2_PO_4_), sodium phosphate dibasic (Na_2_HPO_4_), catalase, human serum, BSA, glucose, galactose, maltose, and 15 different amino acids, including alanine, asparagine, arginine, glycine, histidine, leucine, lysine, methionine, phenylalanine, proline, serine, threonine, tryptophan, tyrosine, and valine, were purchased from Sigma-Aldrich (Milwaukee, WI, USA). AUR was obtained from Invitrogen (Eugene, OR, USA). All solutions were prepared with deionized (DI) water purified by a Milli-Q Purification System (Millipore, Darmstadt, Germany).

### 3.2. Synthesis and Characterization of Nanoflowers

HRP-Cu NFs were synthesized according to the previously reported protocol with minor modifications [20]. Firstly, 60 μL of aqueous CuSO_4_ solution (120 mM) was added to 9 mL of PBS (10 mM, pH 7.4) containing HRP (0.02 mg/mL). The mixture was incubated for 3 days at RT without disturbance. Then, the blue-colored precipitates were collected by centrifugation at 12,000 rpm for 5 min and washed thrice using DI water. BSA-Cu NFs were synthesized by following the same procedures described above but using BSA (0.02 mg/mL) rather than HRP. Copper phosphate (Cu_3_(PO_4_)_2_) precipitate was prepared with the same procedures, but without the involvement of protein.

Morphologies and elemental compositions of HRP-Cu NFs were analyzed by scanning electron microscopy (SEM, Hitachi, S-4700, Tokyo, Japan) with energy dispersive X-ray spectroscopy (EDS, Elementar, Vario Macro, Germany). FT-IR spectra of HRP-Cu NFs were obtained using a FT-IR spectrophotometer (FT/IR-4600, JASCO, Easton, MD, USA) in the spectral range of 500–4000 cm^−^^1^. For XRD analysis (D/MAX-2500, Rigaku Corporation, Tokyo, Japan), scale-up synthetic reactions to 100 mL were performed and the precipitates were collected, washed, and dried at 70 °C for 1 day before XRD measurement. The encapsulation yield of HRP within nanoflowers was calculated from the difference between the initial enzyme amount and the leached amount in the supernatant by using Micro bicinchoninic acid (BCA) protein assay kit (Thermo Fisher Scientific, Waltham, MA, USA) with BSA as a standard.

### 3.3. Detection of Biothiols Using HRP-Cu NFs

Biothiol detection using the dual catalytic activities of HRP-Cu NFs was performed as follows. GSH, Cys, or Hcy (10 μM, 100 μL) was added into a reaction solution containing HRP-Cu NFs (0.1 mg/mL) and AUR (25 μM) in sodium phosphate buffer (10 mM, 700 μL, pH 7.4). After incubation for 15 min at 37 °C, the supernatant was collected by centrifugation (10,000 rpm, 1 min), which was then used to measure the fluorescence intensity with the excitation and emission wavelength of 530 nm and 585 nm, respectively, which correspond to the oxidized AUR. Fluorescence intensities were measured using a Microplate reader (Synergy H1, BioTek, VT, at the Core-facility for Bionano Materials in Gachon University) using black, 384-well Greiner Bio-One microplates (ref: 781077, Courtaboeuf, France). The effects of reaction buffer pH on the cascade reaction activity to detect a selected biothiol (GSH) were investigated following the same procedures, except using sodium phosphate buffer (0.1 M) prepared at various pHs (3.0–10.0). The effects of incubation time, temperature, and the employed concentrations of HRP-Cu NFs and AUR were also evaluated in ranges of time (0–30 min), temperature (4–80 °C), HRP-Cu NFs (0.01–0.6 mg/mL), and AUR (0.1–50 μM). The fluorescence intensities were recorded at each reaction condition.

Stabilities of HRP-Cu NFs and the mixture of BSA-Cu NFs with free HRP were determined by incubating them in ranges of pH (2–12) and temperature (4–80 °C) for 2 h, and their cascade reaction activities for detecting GSH were measured as described above. The activities measured before the incubation were set as 100%, and the relative activity (%) was calculated using the ratio of the residual activity to the initial activity of each sample.

### 3.4. Detection of Biothiols in Human Serum

Human serum was 1000-fold diluted to ascertain its biothiol concentration present in the linear range. Original total biothiol concentration of human serum was determined by a thiol fluorescent detection kit (Thermo Fisher Scientific), and then a predetermined amount of GSH, Cys, or Hcy (0.125, 0.25, and 0.5 μM) was further added into human serum to make spiked samples. Finally, total biothiol concentration of each spiked sample (100 μL) was determined by the same aforementioned procedures. The coefficient of variation (CV (%) = SD/average × 100) and the recovery rate (recovery (%) = measured value/actual value × 100) were determined to evaluate the reproducibility and precision of this assay.

## 4. Conclusions

In conclusion, we developed an innovative strategy to prepare HRP-Cu NFs possessing dual catalytic activities of HRP and biothiol oxidase. The results of investigation demonstrate that the HRP-Cu NFs-based system has high selectivity and sensitivity for the detection of the corresponding biothiols. As the current system enabled rapid detection of target biothiols with sufficiently high stability and detection precision, it should find practical applications in facility limited or POCT environments. To the best of our knowledge, this is the first work to report biothiol oxidase-mimicking activity of hybrid nanoflowers that provides an insight into hybrid nanoflower-type nanozymes that incorporate diverse organic and inorganic substances, as well as broadens the scope of their applications in diagnostic areas.

## Figures and Tables

**Figure 1 ijms-23-00366-f001:**
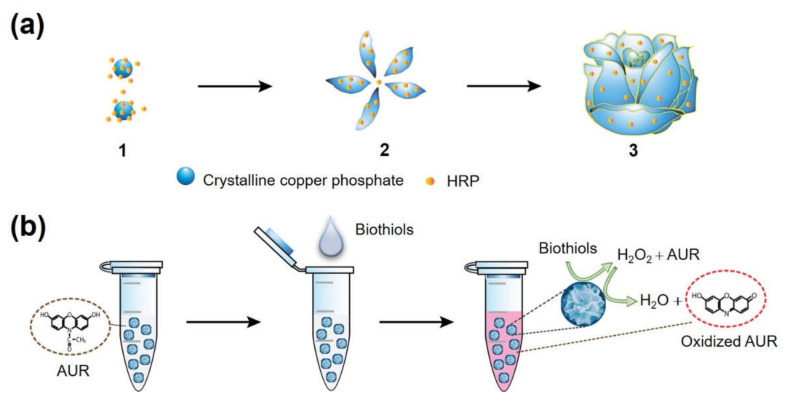
Schematic illustrations for (**a**) the synthesis of HRP-Cu NFs with three stages: (1) nucleation of primary copper phosphate crystals, (2) anisotropic growth of flower petals, and (3) complete formation of the nanoflowers; (**b**) fluorescent one-pot detection of biothiols through the cascade reaction promoted by HRP-Cu NFs.

**Figure 2 ijms-23-00366-f002:**
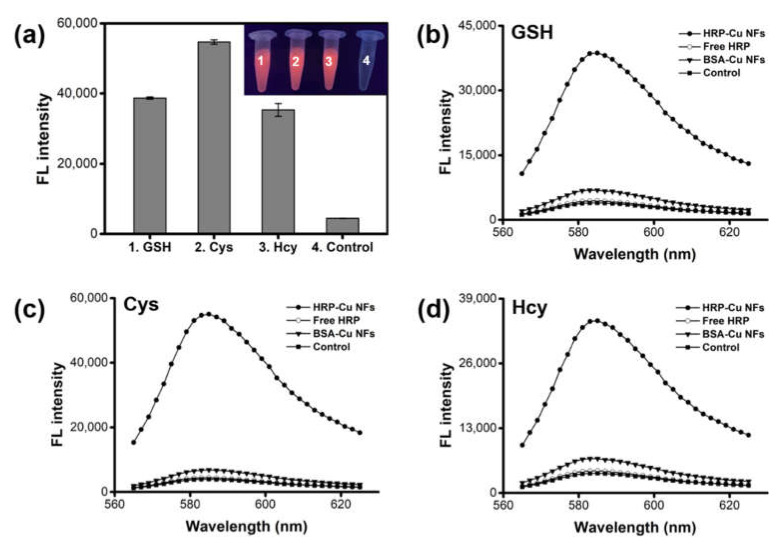
Demonstration for the biothiol detecting activity of HRP-Cu NFs. (**a**) Fluorescence intensities and the corresponding images from the HRP-Cu NFs-based cascade reaction toward GSH, Cys, and Hcy. Control includes HRP-Cu NFs and AUR without biothiols. Fluorescence spectra for the detection of (**b**) GSH, (**c**) Cys, and (**d**) Hcy. The concentration of biothiols was 1 μM.

**Figure 3 ijms-23-00366-f003:**
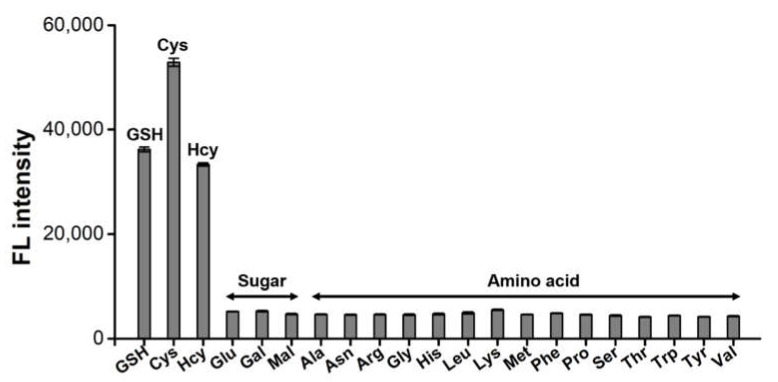
Selectivity of HRP-Cu NFs-based biothiol detection system. 1 μM biothiols (GSH, Cys, and Hcy) as well as 20 μM of other carbohydrates and amino acids were used.

**Figure 4 ijms-23-00366-f004:**
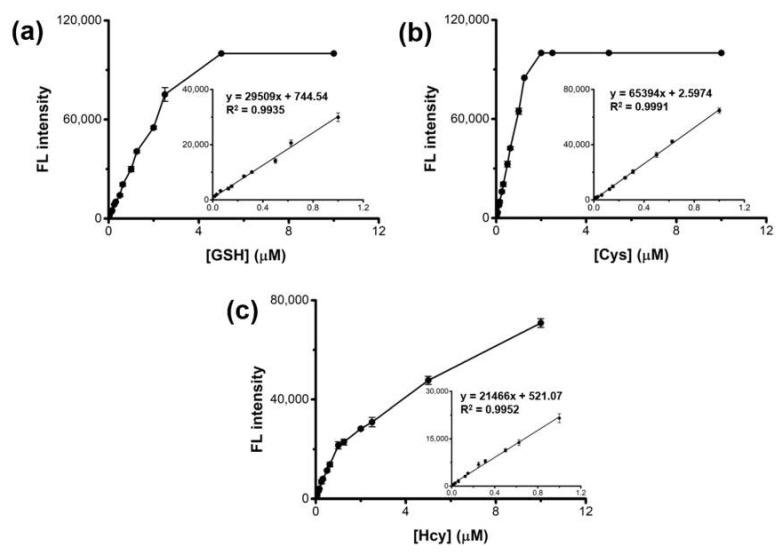
Dose-response curves and the corresponding calibration curves with the linear equations for the detection of (**a**) GSH, (**b**) Cys, and (**c**) Hcy using HRP-Cu NFs-based assay system.

**Table 1 ijms-23-00366-t001:** Determination of biothiols in spiked human serum using HRP-Cu NFs-based assay system.

Original Biothiols (μM)	Added (μM)	GSH			Cys			Hcy		
		Measured (μM)	Recovery (%)	CV (%)	Measured (μM)	Recovery (%)	CV (%)	Measured (μM)	Recovery (%)	CV (%)
0.404	0.125	0.533	100.73	2.17	0.519	98.02	0.11	0.507	95.79	0.59
	0.25	0.639	97.78	1.24	0.665	101.74	0.87	0.653	99.83	0.89
	0.5	0.862	95.35	1.27	0.882	97.57	0.80	0.873	96.53	0.71

## Data Availability

Not applicable.

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
