# Peer review of "Dual-Functional Peroxidase-Copper Phosphate Hybrid Nanoflowers for Sensitive Detection of Biological Thiols"

_ijms, 2021, doi:10.3390/ijms23010366_

Round 1

Reviewer 1 Report

This manuscript by Le et al. described a hybrid nanoflower structure composed of copper phosphate and horseradish peroxidase, which demonstrated highly sensitive detection of several types of biothiol molecules in an in vitro setting. I recommend its publication in IJMS provided that the following comments are addressed.

Major comments:

  1. Line 123-125: if the Cu3(PO4)2 precipitate sample is bulky and not in the nanoflower form, I would attribute the peak intensity difference in XRD to the crystallite size, since the crystal domains in the nanoflowers are expected to be very small. Indeed, the authors should calculate the average crystallite sizes in the nanoflower and the control samples using the Scherrer formula.
  2. The authors “hypothesized” that copper phosphate + thiols can generate H2O2, but they should add references to this. In particular, Cu2+ ions are known to oxidize thiols to generate H2O2. In the case of copper phosphate, is the solid acting as a heterogeneous catalyst, or are the bleached Cu2+ ions into the solution serving as a homogeneous catalyst?

Minor comments:

  1. Make sure all acronyms are defined when they first make a presence.
  2. Line 118-122: it should be clearly conveyed that these results refer to the FTIR data.
  3. Line 123: “JPSCD” should be “JCPDS.”
  4. Line 197: The choice of the equation for LOD should be referenced. Why was 3 chosen as the coefficient in this particular case?

Reviewer 2 Report

The present study “Dual-Functional Peroxidase-Copper Phosphate Hybrid Nanoflowers for Sensitive Detection of Biological Thiols” reports peroxidase containing nanoflower structures for two-step reaction sensing of biothiols. The hybrid nanoflowers composed of HRP and copper phosphate, which has been reported to have peroxidase like activities (e.g. ref 25). These dual enzymatic activities were successfully used to induce a cascade reaction system to detect biothiols. Although used structures are already known and reaction activities are somewhat expected, this method offered nice LOD and detection ranges for biothiols. Therefore, I recommend publication of this work in IJMS after addressing following issues.

  1. Please indicate biothiol concentrations in Fig 2.
  2. Line 157~ ‘HRP-Cu NFs had higher biothiol sensing activity, which was ~1.5-fold higher than that of control system composed of free HRP with BSA-Cu NFs, and ~2-fold higher than that of HRP with Cu3(PO4)2 precipitate (Figure S4).’: please explain 1.5 & 2-fold signal increase by combining HRP and Cu NFs. Is this expected? (higher or lower than expected?)
  3. The biggest concern for this approach is the sensing selective of Cu NFs for biothiols. Although the authors tested amino acids and sugars, there are many other reactive entities in biological samples. Serum samples are, therefore, important to support the value of this work. However, here only 1000-fold diluted serum was used. Is this a standard for serum-spike in experiments? It would be necessary to investigate signals generated by differently diluted serum solutions with and without added biothiols. And then please discuss how to use this method for biologically relevant samples.

Round 2

Reviewer 2 Report

The authors properly revised the manuscript based on the reviewer's comments.